# Oral Frailty as a Risk Factor for Fall Incidents among Community-Dwelling People

**DOI:** 10.3390/geriatrics9020054

**Published:** 2024-04-22

**Authors:** Hisayo Yokoyama, Yugo Kitano

**Affiliations:** 1Research Center for Urban Health and Sports, Osaka Metropolitan University, 3-3-138 Sugimoto, Sumiyoshi-ku, Osaka-shi, Osaka 558-8585, Japan; u5k1tan0.rp@gmail.com; 2Department of Environmental Physiology for Exercise, Graduate School of Medicine, Osaka Metropolitan University, 3-3-138 Sugimoto, Sumiyoshi-ku, Osaka-shi, Osaka 558-8585, Japan

**Keywords:** web-based survey, health app, habitual exercise, older health, fall incidents, oral frailty, primary healthcare

## Abstract

Background: Identifying older adults with a high risk of falling and providing them with appropriate intervention are vital measures for preventing fall incidents. Scholars report that oral frailty, a decline in oral function, is related to physical function; thus, it bears a potential association with fall risks. This study aimed to investigate the relationship between fall incidents and the status of physical and oral frailty among a sample of residents in Osaka Prefecture. Subjects and Methods: This study targeted community-dwelling people aged 50 years and older who responded to an annual questionnaire survey using a health app for 2 consecutive years, namely, 2020 and 2021. This study analyzed responses from 7591 (62 ± 7 years) participants and determined the status of their oral frailty and comprehensive and physical frailty using the Kihon Checklist. Results: In the 2020 and 2021 surveys, 17% and 19% of the participants exhibited oral frailty and experienced a fall in the previous year, respectively. Logistic regression analysis demonstrated that oral frailty (adjusted odds ratio: 1.553) and physical frailty as well as low levels of awareness of frailty were significant explanatory variables of the occurrence of fall incidents during the subsequent year. Conclusions: Future studies are required to elucidate the mechanisms by which oral frailty induces fall incidents.

## 1. Introduction

Fall incidents are a critical public concern, especially in older adults. Falls frequently result in vertebral or hip fractures in older adults with underlying osteoporosis [1,2]. According to the results of epidemiological surveys on hip fractures in Japan, approximately 70% of cases occur in people over 75 years of age, but the incidence increases with age starting at 40 years of age [3,4]. In cases with hip fracture, performing early surgical treatment and encouraging patients to leave the bed as early as possible to maintain preinjury mobility are essential measures [5]. However, if a comorbid disease exists, time is required for perioperative management. For example, in patients with diabetes, poor blood glucose control delays wound healing and increases the risk of infection at the surgical site [6]. Therefore, elective surgery is performed after stabilizing blood glucose levels in advance. Disuse and cognitive dysfunction progress during prolonged periods of bed rest, and new complications, such as pneumonia, occur, which renders impossible the preservation of the quality of life that patients maintained prior to hospitalization [7]. This scenario may result in the increased risk of becoming bedridden and requirement for long-term nursing care.

In addition, Japan is currently facing an unprecedented aging society and shrinking working population. Therefore, the country is promoting the active participation of older adults in the labor market. Consequently, the risk of fall incidents in the workplace is raising concern. In fact, the number of falls at workplaces is increasing, especially among people aged 50 or older, and falls currently account for the largest proportion of industrial accidents (JISHA:Government’s Industrial Accident Prevention Plan [8] https://www.jisha.or.jp/english/govs_plan.html (accessed on 26 February 2024)). Approximately 60% of people that have fallen at workplaces have been forced to take a leave for more than one month, which results in large losses in terms of productivity [9]. The Japanese Ministry of Health, Labour and Welfare is focusing on promoting projects to prevent fall incidents and encouraging employers to not only improve the working environment but also the physical functions of workers and reduce fall risks. Thus, predicting falls and providing appropriate advice about exercise, among others, to them are vital.

*Oral frailty* refers to an accumulation of age-related poor oral health, such as a decline in chewing ability, impaired oral motor skills, and a decreased ability to swallow [10], and is recently attracting scholarly attention. Among the elderly, chewing ability generally decreases due to tooth loss and decreased tongue motor function, which results in oral frailty. Previous studies reported the relationship between the oral and physical function of older adults [11,12,13]. These scholars proposed that chewing ability is an explanatory factor for handgrip strength even after adjusting for whole body muscle mass [14]. In addition, oral function may modulate muscle mass itself via the intake of total energy, protein and vitamin D [15,16]. In fact, there have been reports on the relationship between poor oral function and sarcopenia [16,17]. Furthermore, the loss of natural teeth, inadequate occlusion and impaired mastication increase all-cause mortality [18,19,20]. Although the cause of mortality is not necessarily clarified in these reports, falls are one of the main causes of mortality in older people [21,22] and the decline in lower limb strength and balance has been established as a cause of falls [23]. Therefore, oral frailty may potentially contribute to fall incidents. If this notion is true, then those with oral frailty can be identified as having a high risk of falling, such that measures for preventing fall incidents should be strengthened. However, little research has been conducted about the relationship between fall incidents and oral frailty.

We hypothesized the potential association between the presence of oral frailty and fall incidents. To clarify the hypothesis, we investigated the relationship between fall incidents and the status of physical and oral frailty among a sample of Osaka Prefectural residents.

## 2. Materials and Methods

### 2.1. Participants

The present study targeted community-dwelling people aged 50 years and older who responded to an annual questionnaire survey on frailty using a healthcare smartphone app called “ASMILE” (https://www.asmile.pref.osaka.jp [24] (accessed on 18 January 2024), Figure 1) by Osaka Prefecture for 2 consecutive years, namely, 2020 and 2021. The reason for setting the target age was, as mentioned above, because falls at workplaces mainly occur in people over 50 years of age [8]. People less than 50 years old or who did not answer the items on age, gender, height or weight or who did not complete the responses to the Kihon Checklist (KCL) were excluded. The Research Ethics Committee of the Faculty of Liberal Arts, Sciences and Global Education of Osaka Metropolitan University approved the research protocol (approval number: 2022-09, approved on 1 November 2022). After reading the document that explains research cooperation on a website when registering for an ASMILE account, consent was considered given if the participants ticked the check column.

### 2.2. Procedure

This study was designed as a retrospective cohort study and conducted using the abovementioned web survey via ASMILE from February 2020 to February 2021. The questionnaire comprised the KCL for assessing frailty in addition to questions regarding exercise habits and the awareness of frailty. The time required to answer the questionnaire was approximately 10 min. G*Power was used to calculate the sample size by considering an α-error of 0.05 and a power of 1 − β (0.95). The prevalence of fall incidents and the KCL-based oral frailty among community-dwelling elderly people in Japan were both estimated to be 20% based on previous studies [25,26]. The desired sample size was calculated as 1497. Given the number of people who participate in the survey in typical years, we predicted that the study could obtain a sufficient sample size even if we assumed that participants with incomplete responses would be excluded.

### 2.3. Kihon Checklist

The KCL is a 25-item questionnaire originally developed by the Japanese Ministry of Health, Labour and Welfare to screen elderly people who will need long-term care services in the near future at the welfare counter (https://www.mhlw.go.jp/topics/2009/05/dl/tp0501-1c_0001.pdf [27] accessed on 23 January 2024); see the Appendix A). Therefore, the originally intended target populations for the KCL are different from the target of this study, hence, people over 50 years. However, thus far, the literature has reported that the KCL is closely correlated with the validated assessments of Fried’s frailty phenotypes [28] and that it can be used as a reliable tool for assessing frailty status [29]. In addition, the main outcome of this study was the fall incident, and the KCL is considered to be a questionnaire that can comprehensively investigate the potential contributors to falls. Taking these into consideration, this study adopted the KCL as a web-based questionnaire.

The targeted populations respond to all questions with yes or no. The KCL has seven domains, namely, activities of daily living, physical function, nutrition, oral function, isolation, cognitive function and mood [30]; thus, it can also detect which functions are problematic. Based on previous reports, the present study assessed frailty in specific domains, that is, physical and oral (impaired oral function) and comprehensive frailty using the KCL. Specifically, those who responded “yes” to at least 7 of the 25 items, at least 2 of the 3 questions on oral function (i.e., Do you have any difficulties eating tough foods compared to 6 months ago?, Have you choked on your tea or soup recently? and Do you often experience having a dry mouth?) and at least 3 of the 5 questions on physical activity (Do you normally climb stairs without using handrail or wall for support?, Do you normally stand up from a chair without any aids?, Do you normally walk continuously for 15 min?, Have you experienced a fall in the past year? and Do you have a fear of falling while walking?) in the KCL were classified as comprehensive, oral and physical frailty, respectively [26,29,31].

### 2.4. Other Items

As potential contributors to fall incidents, this study included items on exercise habits and the awareness of frailty in the questionnaire. The participants responded yes or no to the question, “Do you exercise, such as walking, at least once a week?” They also responded to the question, “Do you know the word frailty?” (do not know/have heard the word before/know a little/know well). Height and weight were self-reported (included in Q12 of the KCL) to calculate body mass index (BMI).

### 2.5. Step Counts

ASMILE is linked to the pedometer function of the smartphone, and the app records the daily number of steps. We extracted step count data for 19 days, which spanned the date of responding to the questionnaire, and calculated the average number of steps per day (steps/day).

### 2.6. Data Acquisition from Osaka Prefecture

All abovementioned data were linked to the age and gender information registered in the ASMILE account, which were then provided by Osaka Prefecture in an anonymized state.

### 2.7. Statistical Analysis

This study examined the normal distribution for all variables using the Kolmogorov–Smirnov test and conducted comparisons among the age groups using the chi-square test in the case of categorical variables or a one-way analysis of variance with a post hoc Dunnett’s test in the case of consecutive variables. Comparisons between groups were performed using the chi-square test in the case of categorical variables or using the unpaired *t*-test for consecutive variables. The effect size of the differences between groups was examined using Cohen’s *d* test and classified as small (*d* > 0.20 and <0.50), medium (*d* > 0.50 and <0.80) or large (*d* > 0.80). In addition, it estimated adjusted odds ratios (aORs) and 95% confidence intervals (CIs) for fall incidents via multivariate analysis using binary logistic regression. In the logistic regression analysis, the categorical variables, namely, gender (1 = male, 2 = female), exercise habits (1 = yes, 0 = no), physical frailty (1 = yes, 0 = no), oral frailty (1 = yes, 0 = no) and the awareness of frailty (1 = do not know, 2 = have heard the word before, 3 = know a little, 4 = know well), were coded in the case that they were used as the explanatory variables. Data were analyzed using SPSS 27.0 (IBM Corporation, Armonk, NY, USA). A *p*-value of <5% was considered statistically significant.

## 3. Results

Figure 2 presents a flowchart from participant selection to analysis. A total of 7620 older adults aged 50 years and older responded to the 2020 and 2021 surveys. This study excluded 29 people who provided no responses to the items on age, gender, height or weight or who did not complete the responses to the KCL. Finally, a total of 7591 responses were analyzed.

Table 1 summarizes the responses to the 2020 survey. The prevalence of comprehensive (*p* = 0.005), physical (*p* = 0.002) and oral frailty (*p* < 0.001) was higher in older age groups. The percentage of respondents who reported exercise habits was greater for older age groups (*p* < 0.001); conversely, the percentage of respondents who experienced a fall in the past year was greater for younger age groups (*p* = 0.010). Respondents in older age groups exhibited a better awareness of frailty (*p* < 0.001; Table 1).

Figure 3 depicts the percentages of respondents who experienced a fall in the past year in the 2021 survey. Among them, 18.7% experienced a fall in the past year. The percentages of respondents who experienced a fall in the past year differed among the age groups. For male respondents, the younger age groups reported a higher prevalence of falls (*p* = 0.020; Figure 3).

This study then compared the parameters from the 2020 survey with the respondents with (fall group) and without (no-fall group) the experience of a fall in the past year in the 2021 survey. Table 2 summarizes the results. Respondents in the fall group were younger than those in the no-fall group (*p* = 0.022, effect size Cohen’s d: 0.067). The BMI (*p* = 0.001, effect size Cohen’s d: 0.095) and daily number of steps (*p* < 0.001, effect size Cohen’s d: 0.117) were greater in the fall group than those of the no-fall group. However, the effect sizes were slight in all above cases. The fall group exhibited lower levels of engagement with exercise than the no-fall group (*p* = 0.006); in addition, the level of awareness of frailty was lower in the fall group than the no-fall group (*p* < 0.001). The number of applicable items in the KCL (*p* = 0.022, effect size Cohen’s d: 0.493) and the prevalence of physical (*p* < 0.001) and oral frailty (*p* < 0.001) were greater in the fall group than that of the no-fall group (Table 2). As shown in Figure 4, in all age groups, respondents who had oral frailty in the 2020 survey were more likely to have experienced a fall in the subsequent year than those who did not.

Finally, this study examined the factors associated with fall incidents using logistic regression analysis. The presence of oral frailty (aOR: 1.553; 95% CI: 1.342–1.797) as well as age (aOR: 0.990 per 1 year increase; 95% CI: 0.982–0.998), the awareness of frailty (aORs relative to “do not know” were 0.801 (95% CI: 0.678–0.947), 0.799 (95% CI: 0.675–0.945) and 0.754 (95% CI: 0.627–0.908) in “have heard the word before,” “know a little” and “know well”, respectively) and the presence of physical frailty (aOR: 3.057; 95% CI: 2.514–3.718) were significant explanatory variables for the occurrence of fall incidents during the subsequent year (Table 3).

## 4. Discussion

In this study, the web survey based on the KCL targeted community-dwelling people revealed that the presence of oral frailty was an independent contributing factor to fall incidents over the following year. To the best of our knowledge, the present study is the first to elucidate the relationship between the decline in oral function and fall incidents.

To prevent an increase in the number of patients with fractures due to falls, identifying those at high risk of falling and those at high risk of requiring long-term care, even among elderly people currently living independently, and providing early preventive intervention are crucial measures. In clinical settings, motor function assessment tests such as the Functional Reach Test [32,33,34] and the Timed Up and Go [35] are generally used to assess fall risks among the elderly. In fact, scholars have proposed cutoff values for these test results for identifying people at high risk of falling [32,33,34,35]. However, implementing such motor function assessment tests requires space and manpower for measurement. Furthermore, opportunities for measurement provided for people apart from those who visit the locations where they can take measurements, such as health events and preventive care workshops, are insufficient. The current results imply that by answering only three questions on oral function or five questions on physical strength of the KCL, identifying people with fall risks is possible without any burden on the test taker or examiner.

The literature has reported that oral function declines with the increase in age [36] and that masticatory ability is correlated with entire body motor functions such as grip strength [14,37,38]. Recently, Tagami et al. [39] reported that an increased echogenicity of the tongue, which is a surrogate indicator of tongue motor dysfunction, is associated with frailty assessed using the KCL. In this manner, the relationship between oral function and whole-body motor function has been elucidated. However, even among elderly people who are highly health conscious and observe habitual exercise, they are not necessarily fully conscious of their abilities to chew and swallow anything. Furthermore, in the current study, high levels of awareness of frailty were associated with fewer fall incidents in the subsequent year. This finding indicates that acquiring knowledge about frailty may have resulted in favorable behavioral changes such that frailty and falls are prevented. In this context, it will become increasingly important for older people to improve their health literacy by using smartphone apps like “ASMILE” to acquire useful health information. The simple check tool based on the KCL can be useful for increasing the awareness of care prevention and developing self-management skills among older people. Furthermore, it can be applied as a tool for health guidance and other settings to recommend interventions.

When revised in 2012, the long-term care prevention manual published by the Japanese Ministry of Health, Labour and Welfare emphasized the importance of providing care through a composite care program that includes not only conventional improvement in “musculoskeletal function” but also “nutrition” and “oral function” [40]. Not only poor oral function decline, which makes it difficult to intake all kinds of food including hard meat, but impaired physical strength, making it difficult to shop for ingredients and prepare dishes such as opening the tight lids of bottles, can also lead to the deterioration of nutritional status. Malnutrition leads to a vicious cycle of decreased muscle mass and strength. In other words, these three factors are interrelated, and previous studies propose that providing care in an integrated manner can exert a synergistic effect compared with performing care alone. However, progress in the widespread use of composite care programs in the field of preventive care projects is seemingly insufficient. In many cases, exercise, nutrition, and oral care professionals implement individual programs. In the future, sharing information about a target person among professionals and collaborating to implement care prevention projects are desirable aspects. Furthermore, investigating whether or not interventions for oral frailty reduce fall incidents is crucial.

This study was unable to directly elucidate the mechanism by which poor oral function increases fall risk. As mentioned above, malnutrition is an important mediator between oral frailty and fall risk. There is abundant evidence that poor oral function causes malnutrition and sarcopenia [41,42,43]. Older people sometimes have difficulties not only in eating hard food due to the loss of natural teeth or improper prosthesis but also in swallowing because of insufficient saliva production [42]. An increase in the frequency and time of mastication and a decrease in the sense of taste decrease their appetite itself [42,44]. Consequently, they often suffer from the deficiency of nutrients, especially in macronutrients, which are essential for anabolism and maintaining muscle strength [43]. In addition to mechanisms mediated by malnutrition, physiological functions such as mastication and occlusion appear to directly contribute to maintaining posture and balance. Previous studies examined the relationship between oral function and postural instability. Other studies reported the association of impaired masticatory performance and tooth loss to the decreased ability to recover balance [45,46,47]. Moreover, interventional studies highlighted the improvement in postural and gait stability and muscle strength in the upper and lower limbs after a modulation of the occlusion condition [48]. Scholars also emphasized that teeth clenching can improve flexor reflex and joint fixation in the lower legs while standing [49,50]. Therefore, oral conditions that enable firm chewing may play an important role in maintaining balance and preventing falls. Thus, future studies should elucidate the mechanism through which oral frailty increases fall risk.

Although the presence of exercise habits was not a significant explanatory factor for fall incidents in the logistic regression analysis in the current study, the fall group exhibited low levels of engagement with exercise compared with the no-fall group. On the other hand, the daily number of steps was greater in the fall group than that in the no-fall group. This paradox may be explained by the interpretation that people who stay active in daily life and have a wide range of behaviors obtain more chances of falling during daily activities. In addition, the home and neighborhood environments, such as steps on the road and slippery floors, contribute to fall events [51,52]; therefore, these environmental hazards need to be removed to reduce fall risk for older people [53]. Although the elderly living in rural areas may have a less safe housing environment and, consequently, have a higher incidence and recurrence of falls compared with those living in urban areas [54], the literature has reported that the prevalence of fracture and mortality due to falls are, in fact, lower for the elderly living in rural areas compared with those living in urban areas [55]. The assumed reason underlying this result is that people living in rural areas engage in agricultural work and perform high levels of physical activity [55]. Therefore, to reduce fall risk, not only human factors but also environmental assessment should be considered.

Based on the background that fall incidents at workplaces are occurring especially among people aged 50 or older, this study targeted people over 50 years of age. Contrary to our expectations, the proportion of people who experienced falls is greater in younger age groups in male respondents. According to a survey conducted by the Japanese Cabinet Office, the prevalence of falls is lower in men than in women, but it increases with age, at least for those over 65 years of age [56]. As mentioned above, the discrepancy may be due, at least in part, to the fact that the younger respondents spend more time commuting or working, which affect the activities engaged in prior to falling. A study, which compared the situations of falls in young, middle and older community-dwelling people, reported that most people fell during walking regardless of age group, while young men fell during exercise or running [57]. In addition, previous studies suggested the difference in the etiology of falls between middle-aged and older people. In young and middle-aged adults, factors such as alcohol abuse need to be considered as causes of unintentional falls [58,59]. Finally, middle-aged people generally start to retain various medical issues. Multiple factors other than decreased muscle strength in lower limbs have been identified as causes of falls, especially comorbidity and medications have a significant impact on fall risk [60,61]. Although this study was unfortunately unable to grasp the status of chronic diseases or medications of the respondents, these factors may have modulated the prevalence of falls in the 50–59 years age group both in men and women.

This research has its limitations. First, it used a web survey, and the participants responded using the self-report method. Although the KCL has been proved useful as a scale for evaluating frailty, a discrepancy may exist between the results of the KCL regarding frailty status and the actual results of the motor function test such as muscle strength or gait speed. Similarly, as previously mentioned, this study did not include a nutritional survey, such that malnutrition, among others, may have exerted a mediating effect on the relationship between oral frailty and fall incidents. Furthermore, the subjects are smartphone users who have registered accounts with ASMILE; thus, the study assumes that they are naturally highly conscious about health. A study reported that smartphone use is associated with a greater perceived health status, especially among older adults living in urban areas [62]. It is speculated that a moderate use of smartphones will improve the mental and physical health of older people by encouraging them to connect with the outside world and communicate with others and by making it easier to obtain useful health information. Therefore, directly applying the results to the general population, including those unfamiliar with smartphones or health apps, may be difficult. In the future, a further analysis of factors which can predict fall incidents is needed by conducting surveys using the KCL in actual medical examination settings, where the general population will visit, or by providing options for submitting responses (e.g., filling out a survey form) other than via the web.

## 5. Conclusions

This study conducted a web survey based on the KCL, which targets community-dwelling people to verify the relationship between oral frailty and fall incidents. It found that the presence of oral frailty was an independent contributing factor to fall incidents over the following year. A simple survey using the KCL is useful for identifying people at high risk of falling and is expected to be implemented in health checkups and other settings. In the future, longitudinal studies are required to determine whether or not interventions on oral function, including chewing ability, can improve the nutritional status and musculoskeletal function and reduce the fall risk of older people from the perspective of preventive care. Consequently, evidence acquired from such studies is expected to serve as a reference for the formulation of care plans. Furthermore, feeding back the predicted fall risks as determined using the KCL could be useful for older people in independently managing daily health care.

## Figures and Tables

**Figure 1 geriatrics-09-00054-f001:**
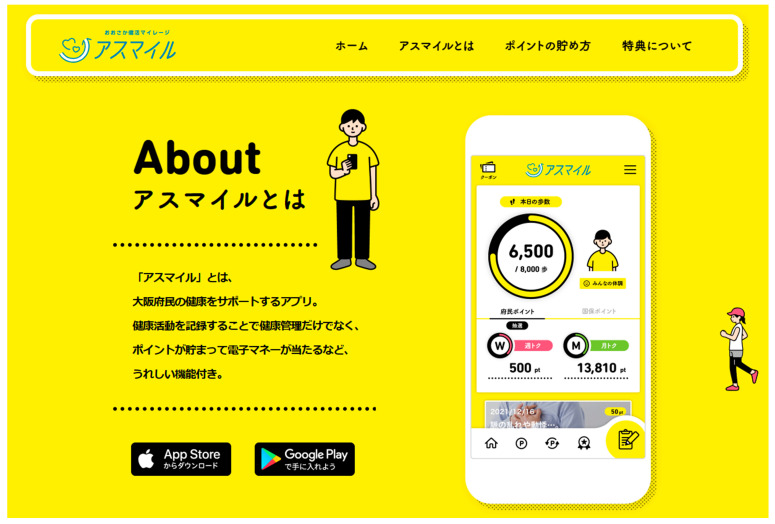
A healthcare smartphone app, “ASMILE” by Osaka Prefecture. “Asmile is an app that not only allows you to manage your health by logging your daily activities, but also gives you incentives such as electronic money”, the explanation in Japanese reads.

**Figure 2 geriatrics-09-00054-f002:**
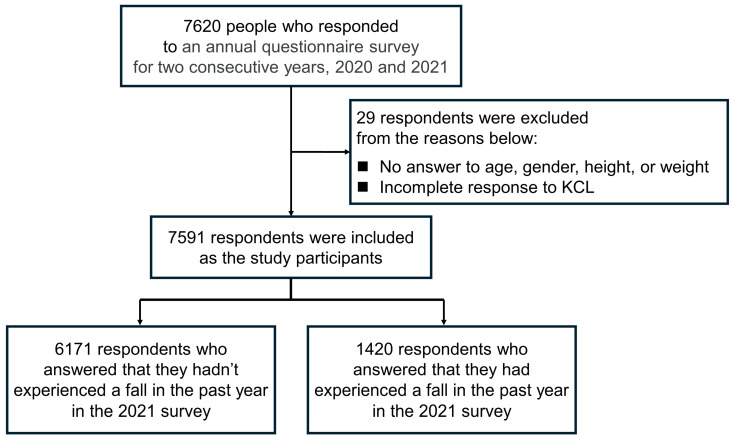
Flowchart from the participants’ selection to analysis.

**Figure 3 geriatrics-09-00054-f003:**
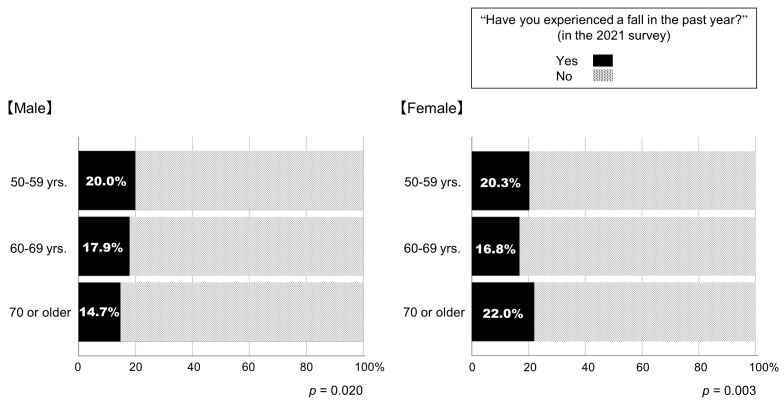
Percentages of participants who experienced a fall in the past year in the 2021 survey.

**Figure 4 geriatrics-09-00054-f004:**
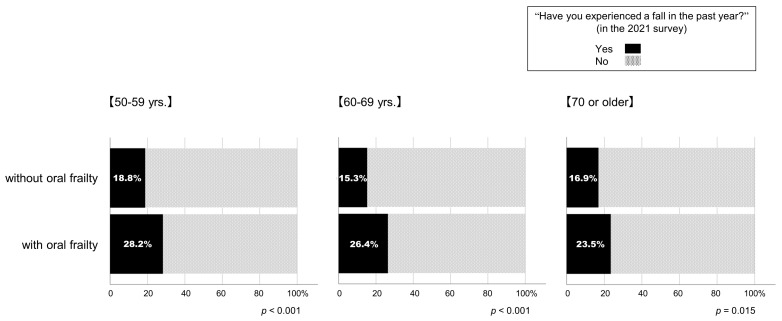
Percentages of participants who experienced a fall in the past year in the 2021 survey according to the presence or absence of oral frailty in the 2020 survey.

**Table 1 geriatrics-09-00054-t001:** Summary of responses and frailty status in 2020 survey.

		Total	50–59 yrs.	60–69 yrs.	70 or Older	*p* Value
n		7591	3224	3096	1271	
Age	(yrs.)	61.6 ± 7.3	54.5 ± 2.8	64.6 ± 2.8 # (95% CI: 9.9–10.2)	72.5 ± 2.7 # (95% CI: 17.7–18.1)	<0.001
	(range)	50–88	50–59	60–69	70–88	
Gender	(male/female)	3021/4570 (39.8%/60.2%)	1121/2103 (34.8%/65.2%)	1247/1849 (40.3%/59.7%)	653/618 (51.4%/48.6%)	<0.001 *
BMI	(kg/m^2^)	22.4 ± 3.1	22.5 ± 3.4	22.4 ± 3.0 (95% CI: −0.3–0.1)	22.4 ± 2.7 (95% CI: −0.3–0.2)	0.377
Number of steps	(steps/day)	6940 ± 4383	6852 ± 4242	6956 ± 4363 (95% CI: −148–355)	7123 ± 4767 (95% CI: −62–603)	0.188
Exercise habits	(yes/no)	6124/1467 (80.7%/19.3%)	2295/929 (71.2%/28.8%)	2675/421 (86.4%/13.6%)	1154/117 (90.8%/9.2%)	<0.001 *
Awareness of frailty	(do not know/have heard the word before /know a little /know well)	3837/1305/1324/1125 (50.5%/17.2%/17.4%/14.8%)	1801/508/507/408 (55.9%/15.8%/15.7%/12.7%)	1505/555/556/480 (48.6%/17.9%/18.0%/15.5%)	531/242/261/237 (41.8%/19.0%/20.5%/18.6%)	<0.001 *
Comprehensive frailty	(yes/no)	2131/5460 (28.1%/71.9%)	967/2257 (30.0%/70.0%)	832/2264 (26.9%/73.1%)	332/939 (26.1%/73.9%)	0.005 *
Physical frailty	(yes/no)	504/7087 (6.6%/93.4%)	194/3030 (6.0%/94.0%)	198/2898 (6.4%/93.6%)	112/1159 (8.8%/91.2%)	0.002 *
Oral frailty	(yes/no)	1275/6316 (16.8%/83.2%)	478/2746 (14.8%/85.2%)	537/2559 (17.3%/82.7%)	260/1011 (20.5%/79.5%)	<0.001 *
Fall incidence in the past year	(yes/no)	1417/6174 (18.7%/81.3%)	651/2573 (20.2%/79.8%)	534/2562 (17.2%/82.8%)	232/1039 (18.3%/81.7%)	0.010 *

BMI, body mass index; CI, confidence interval. Values represent number with percentage or mean ± SD. *: analyzed by chi-square test. # represents the difference vs. the age group 50–59 yrs., and 95% confidence intervals are shown in parentheses.

**Table 2 geriatrics-09-00054-t002:** A comparison of the parameters from the 2020 survey between the respondents who answered they had experienced a fall in the past year and who did not in the 2021 survey.

		Fall Incidents in the Past Year			
		No (No-Fall Group)	Yes (Fall-Group)	*p* Value	Cohen’s *d*
Age	(yrs.)	61.7 ± 7.3	61.2 ± 7.2	0.022	−0.067
Gender	(male/female)	2478/3696	543/874	0.208 *	---
BMI	(kg/m^2^)	22.4 ± 3.1	22.6 ± 3.3	0.001	0.095
Number of steps	(steps/day)	6845 ± 4262	7355 ± 4858	<0.001	0.117
Exercise habits	(yes/no)	5018/1156	1106/311	0.006 *	---
Awareness of frailty	(do not know/have heard the word before/know a little/know well)	3035/1082/1104/953	802/223/220/172	<0.001 *	---
Number of applicable items in KCL	(items/25 items)	4.8 ± 3.0	6.4 ± 3.6	<0.001	0.493
Physical frailty	(yes/no)	294/5880	210/1207	<0.001 *	---
Oral frailty	(yes/no)	937/5237	338/1079	<0.001 *	---

BMI, body mass index; KCL, Kihon Checklist. Values represent number with percentage or mean ± SD. *: analyzed by chi-square test.

**Table 3 geriatrics-09-00054-t003:** Factors associated with fall incidents based on logistic regression analysis.

		aOR	95% CI	*p* Value
Age	yrs.	0.990	0.982–0.998	0.019
Gender (female)	1 = male, 2 = female	1.135	0.998–1.290	0.053
BMI	kg/m^2^	1.018	0.998–1.038	0.073
Exercise habit	0 = no 1 = yes	1.010	0.869–1.174	0.897
Awareness of frailty	1 = do not know 2 = have heard the word before 3 = know a little 4 = know well	(1 as the reference) 0.801 0.799 0.754	0.678–0.947 0.675–0.945 0.627–0.908	0.009 0.009 0.003
Physical frailty	0 = no 1 = yes	3.057	2.514–3.718	<0.001
Oral frailty	0 = no 1 = yes	1.553	1.342–1.797	<0.001

aOR, adjusted odds ratio; CI, confidence interval; BMI, body mass index.

## Data Availability

The data presented in this study are openly available on FigShare at https://figshare.com/articles/dataset/Geriatrics_data_set/25397371, accessed on 2 April 2024.

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
