# Peer review of "Oral Frailty as a Risk Factor for Fall Incidents among Community-Dwelling People"

_geriatrics, 2024, doi:10.3390/geriatrics9020054_

Round 1
Reviewer 1 Report
Comments and Suggestions for Authors
- Table 1 and table 2. Please show percentages for categorical variables.
- Please further discuss the role of malnutrition as a mediating factor for the development of frailty and fall risk in older adults
- Given that this is an app-based study, is there any evidence on the relationship between digital literacy and falls, physical performance, or oral health in older adults?
Author Response
We appreciate Reviewer 1 for the kind and important comments. Our point-by-point responses are as below.
Table 1 and table 2. Please show percentages for categorical variables.
According to Reviewer 1's advice, to aid the reader's understanding, we have added percentages to the categorical variables in Tables 1 and 2.
Please further discuss the role of malnutrition as a mediating factor for the development of frailty and fall risk in older adults
We expanded the discussion about the role of malnutrition as a mediator in the discussion section, especially in the 5th paragraph.
- Given that this is an app-based study, is there any evidence on the relationship between digital literacy and falls, physical performance, or oral health in older adults?
Thank you for your important comments. Research has shown that elderly people who use smart devices show a higher level of perceived health. Taking this into account, we have revised the limitation section.
Reviewer 2 Report
Comments and Suggestions for Authors
Thank you for your work and for the opportunity to review this paper.
The topic of the article seems interesting to me, and I believe they have a sufficient sample size to conduct an engaging analysis. However, I consider that in the current form of the article, the results do not align with the hypothesis of their study, making it difficult to justify in its present form. I advise the authors to repeat the analysis, changing the approach according to their hypotheses, which could make the article more interesting and methodologically sound.
To aid the reader, it might be beneficial if the abstract were divided into introduction, materials and methods, results, and conclusions.
It would be advisable for the authors to review their Keywords.
The sample is too young to be considered elderly; perhaps "older healthy" would be more appropriate. On the other hand, it would be interesting to add "falls" and "oral frailty."
This sentence may be controversial: "However, if a comorbid disease exists, such as diabetes, then additional time is required for perioperative management." I advise the authors to either substantiate this claim with literature, as it currently lacks a bibliographic citation to justify it.
I advise the authors to revise the introduction, as it is currently phrased, they make generalizations that do not necessarily apply to all cases. However, they write these premises as categorical assertions that do not always have to be met.
In Figure 1, there is text in Japanese; it would be advisable for the authors to translate this into English, as the reader cannot understand the information in the figure.
It would be advisable for the authors to conduct a multivariate analysis adjusted for the main confounding factors, such as age, sex, comorbidity, physical frailty, etc. In Table 2, one can observe the variables that show differences between both groups and may influence the result. Other variables such as the number of medications and polypharmacy are missing, which are necessary to understand the sample better and affect the results. I advise the authors to review the current literature to know the risk factors for falls, the following article is interesting in this regard: Montero-Odasso, Manuel, et al. "World guidelines for falls prevention and management for older adults: a global initiative." Age and ageing 51.9 (2022): afac205.
In Table 1, the authors divide the results by age groups, but their study is really about oral frailty. It would be more appropriate if they were divided into participants with oral frailty and without oral frailty.
The same occurs in Figure 3, where they divide it by sex instead of by oral frailty.
Author Response
We appreciate Reviewer 2 for the kind and important comments. Our point-by-point responses are as below.
To aid the reader, it might be beneficial if the abstract were divided into introduction, materials and methods, results, and conclusions.
According to the suggestion, we added subheadings to the abstract.
It would be advisable for the authors to review their Keywords.
The sample is too young to be considered elderly; perhaps "older healthy" would be more appropriate. On the other hand, it would be interesting to add "falls" and "oral frailty."
Thank you for the advice on the Keywords. We revised and added some keywords following your suggestion.
This sentence may be controversial: "However, if a comorbid disease exists, such as diabetes, then additional time is required for perioperative management." I advise the authors to either substantiate this claim with literature, as it currently lacks a bibliographic citation to justify it.
Taking your point into consideration, I have revised the expressions that could be seen as controversial and added a new quotation.
I advise the authors to revise the introduction, as it is currently phrased, they make generalizations that do not necessarily apply to all cases. However, they write these premises as categorical assertions that do not always have to be met.
According to the suggestion, we revised the introduction section.
In Figure 1, there is text in Japanese; it would be advisable for the authors to translate this into English, as the reader cannot understand the information in the figure.
Thank you for your advice. We included the translation of the Japanese text in the figure legend.
It would be advisable for the authors to conduct a multivariate analysis adjusted for the main confounding factors, such as age, sex, comorbidity, physical frailty, etc. In Table 2, one can observe the variables that show differences between both groups and may influence the result. Other variables such as the number of medications and polypharmacy are missing, which are necessary to understand the sample better and affect the results. I advise the authors to review the current literature to know the risk factors for falls, the following article is interesting in this regard: Montero-Odasso, Manuel, et al. "World guidelines for falls prevention and management for older adults: a global initiative." Age and ageing 51.9 (2022): afac205.
As shown in Table 2, we found contradictory results in the fall-group, with a smaller proportion of people taking exercise habits, but a higher number of steps. Therefore, in the logistic regression analysis, only the presence or absence of exercise habits was included as an explanatory variable. Additionally, as you pointed out, many factors other than decreased muscle strength in lower limb have been identified as causes of falls. Although, unfortunately, we were not able to investigate polypharmacy etc. in this study, we have mentioned these topics in a new paragraph of the discussion section.
In Table 1, the authors divide the results by age groups, but their study is really about oral frailty. It would be more appropriate if they were divided into participants with oral frailty and without oral frailty.
The same occurs in Figure 3, where they divide it by sex instead of by oral frailty.
Thank you for your advice. Although this study is all about oral frailty as you pointed out, the main outcome was fall incident. Therefore, we aimed to investigate oral frailty as a role of contributor to fall incident. For this reason, we would like to leave Table 1 and Figure 3 as original. However, we have added a new Figure 4 to visually illustrate the impact of oral frailty on the prevalence of falls.
Round 2
Reviewer 2 Report
Comments and Suggestions for Authors
The article has been substantially improved and is ready for its publication.